# An equation of state unifies diversity, productivity, abundance and biomass

John Harte [1,2,3 ✉], Micah Brush[1,4], Erica A. Newman [5] & Kaito Umemura[1,6]

To advance understanding of biodiversity and ecosystem function, ecologists seek widely applicable relationships among species diversity and other ecosystem characteristics such as species productivity, biomass, and abundance. These metrics vary widely across ecosystems and no relationship among any combination of them that is valid across habitats, taxa, and spatial scales, has heretofore been found. Here we derive such a relationship, an equation of state, among species richness, energy flow, biomass, and abundance by combining results from the Maximum Entropy Theory of Ecology and the Metabolic Theory of Ecology. It accurately captures the relationship among these state variables in 42 data sets, including vegetation and arthropod communities, that span a wide variety of spatial scales and habitats. The success of our ecological equation of state opens opportunities for estimating difficult-to-measure state variables from measurements of others, adds support for two current theories in ecology, and is a step toward unification in ecology.

[1] The Energy and Resources Group, University of California, Berkeley, CA 94720, USA. [2] The Rocky Mountain Biological Laboratory, Gothic, CO 81224, USA. [3] The Santa Fe Institute, Santa Fe, NM 87501, USA. [4] The Department of Mathematical and Statistical Sciences, University of Alberta, Edmonton, AB T6G 2G1, Canada. [5] School of Natural Resources & the Environment, University of Arizona, Tucson, AZ 85721, USA. [6] Graduate School of Human Development and Environment, Kobe University, Kobe, Hyogo 657-8501, Japan. ✉email: jharte@berkeley.edu

A major focus of ecology is the study of species diversity across ecosystems and its relationship to ecosystem structure and function[1–8]. Considerable effort has been directed at the challenge of finding other macro-level ecosystem variables, such as productivity, that might correlate with and potentially explain the wide range of values of species richness observed in different habitats, climates, and taxa across spatial scales. Species diversity and productivity are system-level descriptors of ecosystems. Other system descriptors include the total biomass and abundance of individuals.

Diversity, productivity, abundance, and biomass in ecology are loosely analogous to state variables in thermodynamics, such as the pressure, volume, temperature, and the number of moles of a container of gas. In thermodynamics, a universal relationship among state variables, also known as an equation of state, exists in the form of the ideal gas law: $PV = nRT$. Equations of state are common in physics and chemistry and derive from fundamental theory, but in macroecological studies of ecosystems, such framing has been lacking. A successful equation of state derived from ecological theory would deepen our understanding of ecology, allow prediction of diversity or productivity from knowledge of other system-level state variables, and potentially enhance applications of ecological theory to conservation and restoration[9].

In thermodynamics, it has proven useful to distinguish the micro-level and the macro-level descriptions of the system and then maximize Shannon information entropy[10,11] to infer phenomena at the micro-level from constraints imposed by state variables at the macro-level. For example, the Boltzmann distribution of molecular kinetic energies can be derived from knowledge of the total system energy and the number of molecules. Extending this concept to ecology, we take the micro-level variables to be the metabolic rates, $\varepsilon$, of individuals and the abundances, $n$, of species within an ecological community of, for example, plants, arthropods, or mammals. We take the macro-level state variables to be the total number of species, $S$, the total number of individuals, $N$, in the community, and the total metabolic rate, $E$, of all the individuals in a given area $A$. An application of MaxEnt then results in the Maximum Entropy Theory of Ecology (METE)[12–14], which we use to derive an equation of state.

At the core of METE is the "ecosystem structure function" $R(n, \varepsilon | S, N, E)$, a joint probability distribution over two micro-level variables, abundance $n$, and metabolism $\varepsilon$, that is constrained by values of the state variables $S$, $N$, and $E$. $R \cdot d\varepsilon$ is the probability that if a species is picked from the species pool, then it has abundance $n$, and if an individual is picked at random from all the species with abundance $n$, then its metabolic energy requirement is in the interval $(\varepsilon, \varepsilon + d\varepsilon)$. From the definition of $R$, total abundance, $N$, is $S$ times the average of $n$, and total metabolic rate, $E$, is $S$ times the average of $n\varepsilon$, where both averages are taken over the distribution $R$. The form of $R$ is derived by maximizing its Shannon information entropy subject to the constraints imposed by the two independent ratios that can be formed from $S$, $N$, and $E$: $N/S$ and $E/S$. The MaxEnt solution[12] is $R = e^{-\lambda_1 n} e^{-\lambda_2 n\varepsilon}/Z$ where the $\lambda_i$ are Lagrange multipliers and $Z$ is the normalization factor, all of which depend only on $S$, $N$, and $E$.

From the ecosystem structure function, predictions for the distribution of abundances across species and the distribution of metabolic rates across individuals, as well as a relationship between the abundance of a species and the average metabolism of its individuals, can be derived[12–14]. In spatially explicit applications, the area of the system is included as a fourth state variable and additional predictions follow, including a universal scale-collapse expression for the species-area relationship[15]. If higher taxonomic levels such as genus or family are included as additional state variables in addition to species, then the theory predicts the distribution of species over these added taxonomic categories and explicit dependence of the abundance-metabolism relationship on the taxonomic structure of the community[16].

Across ecosystems, no one of the three state variables, $S$, $N$, and $E$, that define METE at the macro-level is accurately predicted by the other two. Although models have been proposed that do relate species richness to the abundance or to either productivity or metabolic rate[6,17,18], empirically, there is considerable scatter around such relationships (Supplementary Note 1 and Supplementary Video). In a pioneering study[6], Fisher et al. derived a parameterized relationship between $S$ and $N$ from the widely-applicable log-series distribution of abundances. However, the parameter (Fisher's alpha) is observed to vary considerably from one ecosystem to another, from one taxonomic group to another, and across spatial scales. That variability has not been shown, either theoretically or empirically, to be explainable by abundance or productivity[17].

Various other semi-empirical relationships among biomass, species richness, abundance, and productivity or metabolic rate have been proposed as well[5,7,8]. Moreover, an effort has been directed at linking patterns in macroecology by searching for associations among hypothesized power-law exponents used to characterize various scaling relationships among, for example, abundance, body size, and spatial distribution[19,20]. None of these efforts have yielded the sought-after widely-applicable unification of state variables in ecology.

Here we derive an equation of state for ecology by combining results from the maximum entropy theory of ecology[12–14] with a mass-metabolism scaling rule from the metabolic theory of ecology[18,21]. The equation of state provides a relationship among four state variables: total biomass, $B$, total metabolic rate, $E$, the total number of individuals, $N$, and species richness, $S$. We demonstrate the accuracy of this equation of state across a wide variety of taxa, habitats, and spatial scales.

## Results

To derive the relationship among macro-level ecological variables, which would constitute an ecological analog of the thermodynamic equation of state, we introduce a fourth state variable, $B$, the total biomass in the community. The ecological analog of the thermodynamic equation of state, an expression for biomass, $B$, in terms of $S$, $N$, and $E$, arises if we combine METE with a scaling result from the metabolic theory of ecology (MTE)[18,21]. In particular, we assume the MTE scaling relationship between the metabolic rate, $\varepsilon$, of an individual organism and its mass, $m$: $\varepsilon \sim m^{3/4}$. Without loss of generality[22], units are normalized such that the smallest mass and the smallest metabolic rate within a censused plot are each assigned a value of 1. With this units convention, the proportionality constant in this scaling relationship can be assigned a value of 1. From the definition of the structure-function, it follows[23] that averaging the biomass of individuals times the abundance of species, $n\varepsilon^{4/3}$, over the distribution $R$ and multiplying by the number of species gives the total ecosystem biomass as a function of $S$, $N$, and $E$. Explicitly:

$$B = S \sum_n n \int d\varepsilon\, \varepsilon^{4/3} R(n, \varepsilon | S, N, E) \qquad (1)$$

Both the sum and integral in the above equation can be calculated numerically, and Python code to do so for a given set of state variables $S$, $N$, and $E$, is available at github.com/micbru/equation of_ state/.

We can also approximate the solution to Eq. 1 analytically (Supplementary Note 2) to reveal the predicted functional

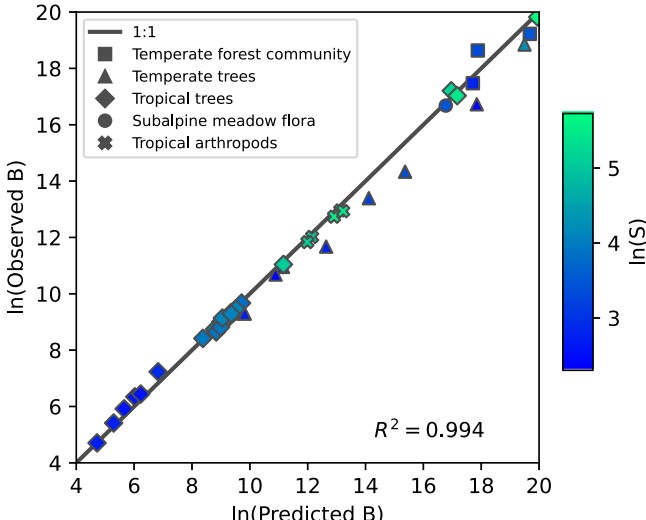

**Fig. 1 A test of the ecological equation of state.** Observed biomass is determined by either summing empirical masses of individuals or summing empirical metabolic rates raised to the ¾ power of each individual. Predicted biomass is determined from Eq. 1 using observed values of $S$, $N$, and $E$. The quantity ln(predicted biomass) explains 99.4% of the variance in observed biomass. Units of mass and metabolism are chosen such that the masses of the smallest individuals in each dataset are set to 1 and those individuals are also assigned a metabolic rate of 1. The shape of the marker indicates the type of data, and the lighter color corresponds to higher species richness. Data for all analyses come from tropical trees[39–45], temperate trees[30–33,46–48], temperate forest communities[27,49], subalpine meadow flora[28], and tropical island arthropods[50].

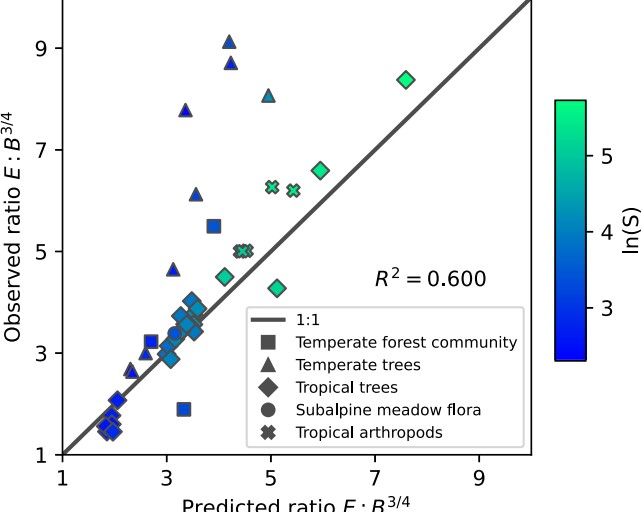

**Fig. 2 The explanatory power of diversity and abundance.** The observed ratio $E/B^{3/4}$ is plotted against the ratio predicted by Eq. 1. Of the fourfold variability across ecosystems in that ratio, 60% is explained by the variability in the predicted combination of diversity and abundance. The shape of the marker indicates the type of data, and the lighter color corresponds to higher species richness. Data for all analyses come from tropical trees[39–45], temperate trees[30–33,46–48], temperate forest communities[27,49], subalpine meadow flora[28], and tropical island arthropods[50].

relationship among the four state variables. If $E \gg N \gg S \gg 1$:

$$B = c\frac{E^{4/3}}{S^{1/3}\ln(1/\beta)} \qquad (2)$$

where $c \approx (7/2)\Gamma(7/3) \approx 4.17$ and $\beta = \lambda_1 + \lambda_2$ is estimated[13,22] from the relationship $\beta\ln(1/\beta) \approx S/N$. Equation 2 approximates the numerical result to within 10% for 5 of the 42 datasets analyzed here, corresponding to $N/S$ greater than ~100 and $E/N$ greater than ~25. Multiplying the right-hand side of Eq. 2 by $1 - 1.16\beta^{1/3}$ approximates the numerical result to within 10% for 33 of the 42 datasets analyzed here, corresponding to $N/S$ greater than ~3 and $E/N$ greater than ~5. The inequality requirements are not necessary for the numerical solution of Eq. 1, which is what is used below to test the prediction.

Empirical values of $E$ and $B$ can be estimated from the same data. In particular, if measured metabolic rates of the individuals are denoted by $\varepsilon_i$, where $i$ runs from 1 to $N$, then $E$ is given by the sum over the $\varepsilon_i$ and $B$ is given by the sum over the $\varepsilon_i^{4/3}$. Similarly, if the mass, $m_i$, of each individual is measured, then $B$ is the sum over the $m_i$ and $E$ is the sum over the $m_i^{3/4}$. In practice, for animal data, metabolic rate is often estimated by measuring mass and then using metabolic scaling, while for tree data, metabolic rate is estimated from measurements of individual tree basal areas, which are estimators[5] of the $\varepsilon_i$.

With $E$ and $B$ estimated from the same measurements, the question naturally arises as to whether a simple mathematical relationship holds between them, such as $E = B^{3/4}$. If all the measured $m$'s, are identical, then all the calculated individual $\varepsilon$'s are identical, and with our units convention we would have $E = B$. More generally, with variation in masses and metabolic rates, the only purely mathematical relationship we can write is inequality between $E$ and $B^{3/4}$: $E = \sum\varepsilon_i \geq (\sum\varepsilon_i^{4/3})^{3/4} = B^{3/4}$. Our derived equation of state (Eq. 2) can be interpreted as expressing the

theoretical prediction for the quantitative degree of inequality between $E$ and $B^{3/4}$ as a function of $S$ and $N$.

A test of Eq. 1 that compares observed and predicted values of biomass with data from 42 censused plots across a variety of habitats, spatial scales, and taxa is shown in Fig. 1. The 42 plots are listed and described in Table S2 and Supplementary Note 3. The communities censused include arthropods and plants, the habitats include both temperate and tropical, and the census plots range in area from 0.0064 to 50 ha. As seen in the figure, 99.4% of the variance in the observed values of $B$ is explained by the predicted values of $B$.

Figure 2 addresses the possible concern that the success of Eq. 1 shown in Fig. 1 might simply reflect an approximate constancy, across all the datasets, of the ratio of $E$ to $B^{3/4}$. If that ratio were constant, then $S$ and $N$ would play no effective role in the equation of state. Equation 1 predicts that variation in the ratio depends on S and N in the approximate combination $S^{1/4}\ln^{3/4}(1/\beta(N/S))$. In Fig. 2, the observed and predicted values of $E/B^{3/4}$ calculated from Eq. 1, are compared, showing a nearly fourfold variation in that ratio across the datasets. The equation of state predicts 60% of the variance in the ratio.

Figure 3 shows the dependence on $S$ and $N$ of the predicted ratio $E/B^{3/4}$ over empirically observed values of $S$, $N$, and $E$. We examined the case in which $S$ is varied for two different fixed values of each of $N$ and $E$ (Fig. 3a) and $N$ is varied for two different fixed values of $S$ and $E$ (Fig. 3b). The value of $E$ does not have a large impact on the predicted ratio, particularly when $E \gg N$. On the other hand, the predicted ratio depends more strongly on $N$ and $S$.

The total productivity of an ecological community is a focus of interest in ecology[1], as a possible predictor of species diversity[24] and more generally as a measure of ecosystem functioning[25]. By combining the METE and MTE frameworks, we can now generate explicit predictions for certain debated ecological relationships, including one between productivity and diversity. Interpreting total metabolic rate $E$ in our theory as gross productivity, then in the limit $1 \ll S \ll N \ll E$, we can rewrite Eq. 2

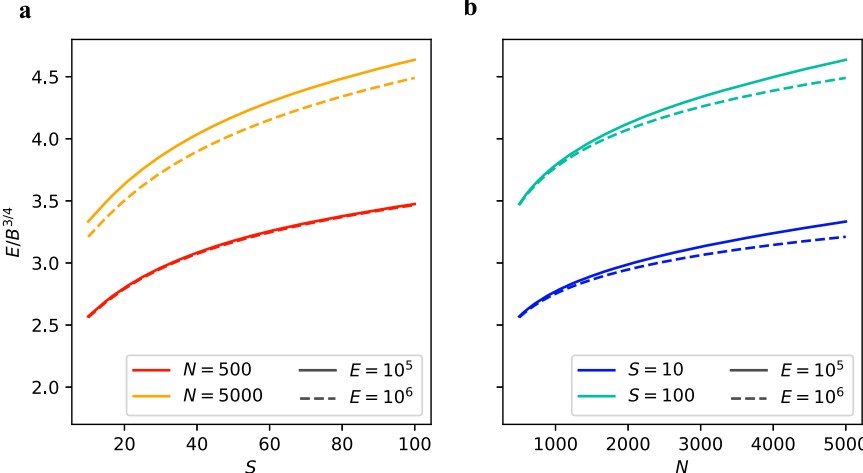

**Fig. 3 The theoretical prediction for the ratio $E/B^{3/4}$ as a function of $S$ and $N$.** The biomass $B$ is predicted by holding $E$ fixed along with one other state variable. In **a** $N$ is fixed and $S$ is varied, and in **b** $S$ is fixed and $N$ is varied. The fixed values are chosen to be roughly consistent within a range of the data considered. The color of the lines represents the corresponding fixed value of $N$ or $S$, while the solid and dashed lines represent different fixed values of $E$.

to highlight the role of diversity and the other state variables in determining this quantity:

$$E = c^{-3/4}S^{1/4}B^{3/4}\ln^{3/4}(1/\beta) \quad (3)$$

here, as shown in Eq. 2, $c$ is a universal constant ($\sim 4.17$). From Eq. 3, and the fact that up to a logarithmic correction $\beta$ varies as $S/N$, biomass exerts the strongest influence on $E$. With biomass fixed, productivity varies approximately as $S^{1/4}$, with a logarithmic correction. With biomass and species richness fixed, the dependence of productivity on abundance is logarithmic. Thus, the dependence of productivity on abundance is weakest, on biomass is strongest, and on species richness is of intermediate strength. Testing these predictions requires finding communities with the same values of pairs of state variables, and thus it is most feasible to compile further tests of Eq. 1 by allowing all four variables to vary as in Figs. 1 and 2. If $N \gg S$, then $\beta$ is a function of $N/S$ and we can re-express Eq. 3 entirely in terms of three ratios, $E/S$, $B/S$, and $N/S$, rather than the four state variables, and obtain:

$$E/S = c^{-\frac{3}{4}}\left(\frac{B}{S}\right)^{\frac{3}{4}}\ln^{\frac{3}{4}}\left(1/\beta\left(\frac{N}{S}\right)\right) \quad (4)$$

To date, empirical cross-ecosystem surveys of relationships among community metabolism, biomass, species richness, and abundance have largely focused on uncontrolled pairwise relationships as, for example[1], between productivity and biomass but without controlling for species richness and abundance, and found considerably more scatter in the relationship than is shown in Fig. 1. In Supplementary Note 1, pairwise comparisons among state variables are shown for the datasets used here, illustrating the absence of strong relationships among pairs of state variables when the remaining pair of state variables is unconstrained. The largest $R^2$ value in all comparisons is 0.987, between $\ln(B)$ and $\ln(E)$, which is expected as they are computed from the same data. Noting $(1 - 0.994)/(1 - 0.987) \approx 0.46$, the unexplained variance in the observed $\ln(B)$ using the equation of state is less than half that of the second-best predictor, $\ln(E)$. All other tested relationships shown in Supplementary Note 1 were significantly worse.

## Discussion

Far more testing of the ecological equation of state is warranted. Moreover, Eq. 1 raises many new questions. As with other empirical successes of METE, it is unclear why an apparently mechanism-free theory should work at all in ecology. Our tentative answer is that in ecosystems in which the state variables are relatively constant in time, the myriad of mechanisms and traits, differing from organism to organism and species to species, confer upon them all sufficient fitness to co-exist[14,26]. In other words, just as in an ideal gas where interactions among molecules can be ignored in thermodynamic equilibrium except under extreme conditions, the mechanisms operating in ecosystems can be ignored because they simply represent different tactics for achieving the common goal of sufficient fitness.

Pursuing that idea, we might more generally expect the predictions of a top-down MaxEnt approach to fail under ecological disturbances that sufficiently alter the fitness landscape, resulting in dynamic, not static, state variables. This indeed appears to be the case for METE's prediction of the species-abundance distribution (SAD) and the species-area relationship (SAR), as discussed elsewhere[14,23,27–29]. An implication of this is that a survey of highly disturbed ecosystems with rapidly-changing state variables might reveal significant deviations from Eq. 1. As in thermodynamics, where a failure of the ideal gas law under extreme values of the state variables revealed the existence of van der Waals forces between polar molecules, so different types of failure of Eq. 1 in ecosystems with rapidly changing state variables might shed light on the causes of disturbance[23].

Applying those insights to our equation of state, four data points stand out as outliers in Fig. 1 and especially Fig. 2, These are the four temperate tree communities (blue triangles). One of the common characteristics of these four sites is that they are currently undergoing secondary succession processes from disturbances over the past decades to centuries[30–33]. Moreover, they stand out from the other sites of their relatively small $S$-to-$N$ ratio (Supplementary Note 3). A more thorough survey of the validity of Eq. 1 across a spectrum of levels and kinds of natural and anthropogenic disturbance is suggested.

We note that the ecosystem area does not appear explicitly in Eq. 1; the area only enters implicitly through the area-dependence of the state variables. At least over the range of areas spanned by our datasets (50 ha/0.0064 ha $\sim 10^4$), this absence of explicit area-dependence in our equation of state is validated. However, just as some of METE's other predictions fail at large spatial scales encompassing multiple ecoregions[13,14], we expect to see significant deviations from the predicted equation of state at very large scales.

The full mass-metabolism scaling relationship in MTE[18] includes a temperature-dependent multiplicative term such that $\varepsilon \sim e^{-E_0/kT} m^{3/4}$, where $E_0$ is the activation energy, $k$ is Boltzmann's constant, and $T$ is the temperature in Kelvin. The effect of this temperature correction on the accuracy of the equation of state remains to be tested.

We have used a metabolic scaling exponent of 3/4 in all of the above, but there is both controversy over, and empirical variability in, its actual value[34]. If the scaling exponent is taken to be 2/3, as can be obtained from an energy budget model in which energy loss is proportional to the surface area of an organism[35] then the exponent of 4/3 in Eq. 1 becomes 3/2, and a different form of the equation of state is obtained. Moreover, the empirical value of $E$ calculated from measured values of the masses of individuals will differ, and thus the empirical ratio of $E$ to $B$ will differ. In Supplementary Note 4 we show that the empirical validity of the equation of state is reduced if the metabolic scaling exponent is assumed to be 2/3 instead of 3/4. In particular, for the test shown in Fig. 1, the $R^2$ value drops from 0.994 to 0.986, and for Fig. 2, the $R^2$ drops from 0.600 to 0.477. On the other hand, if the metabolic scaling exponent is 1, then with our units convention, empirically, $E = B$ and Eq. 1 predicts exactly that. An example of a community in which $E \approx B$ might be a microbiome in which the masses of bacteria differ from one another by a much smaller factor than, for example, the masses of trees in a forest differ from one another. It is noteworthy that for microorganisms, there is evidence[18] that the metabolic scaling exponent is, in fact, closer to 1 than to 3/4.

Ecosystems exhibit numerous idiosyncratic phenomena, but ubiquitous patterns nevertheless exist. The latter motivates the search for general laws. We have provided evidence here for the validity of one such law, an ecological equation of state, which can be derived by combining two ecological theories, METE and MTE. Each of these theories had previously been shown to have broad explanatory power, and our result demonstrates the utility of combining theories that in combination yield more than the sum of the parts[36]. Our ecological equation of state is a simple property of complex ecosystems, and it appears to be valid across spatial scales, across types of habitat, and across different taxonomic groups. Parallel advances in our understanding of relationships among macro-scale variables in other types of complex systems, such as economies[37] or networks[38] may also be possible by combining the powerful MaxEnt inference procedure with appropriate scaling laws.

## Methods

**Data curation and processing**. We searched several databases for data representing censused, species-level information for organisms from a well-defined area, such as a plot or single tree canopy, which included the size of individuals and abundances. We selected data sources in which at least ten species were represented due to the prediction accuracy constraints within METE[22]. Datasets were further investigated through metadata and associated publications. Any datasets that had a recent history of major natural disturbance or human alterations, such as logging and roadbuilding activities, were excluded. After identifying suitable candidate datasets, all data were then processed in the R programming language using a similar workflow.

For plant data, records were filtered to include only live individuals with size measurements. Where individual stem diameters were recorded, we grouped them by individual tree. Diameter at breast height (DBH) measurements were then combined using $\text{DBH}_{\text{new}} = \sqrt{\text{DBH}_1^2 + \ldots + \text{DBH}_n^2}$. For plants, measurements of a size such as DBH and leaf area are considered proxies for the metabolic rates of those individuals. Plant data were processed such that the smallest individual metabolic rate measurement was rescaled to $\varepsilon_{\min} = 1$, and $B = \sum_1^N \text{metabolic rate}_{i,\text{rescaled}}^{4/3}$ (for the main manuscript) or $B = \sum_1^N \text{metabolic rate}_{i,\text{rescaled}}^{3/2}$ (Supplementary Note 4), depending on the scaling relationship being tested.

One modification was made for calculating $E$ for the Point Reyes datasets. For both plots, only the sizes of the largest individuals (trees) were measured. Because the value of $E$ depends almost entirely on the large individuals, we estimated the

size of the smallest individual from photographs for rescaling and employed those values for subsequent calculations.

Animal data is processed similarly to plant data, with the change that measurements of mass are directly measuring biomass, and therefore the metabolic rates are the calculated quantities. The individual with the smallest mass was therefore rescaled to $\varepsilon_{\min} = 1$; then all masses were rescaled using this convention. Rescaled masses were then summed to calculate $B$, and $E$ was calculated using ¾ or 2/3 scaling, such that $E = \sum_1^N \text{mass}_{i,\text{rescaled}}^{3/4}$ or $E = \sum_1^N \text{mass}_{i,\text{rescaled}}^{2/3}$.

**Statistics and reproducibility**. Data were processed in R. All statistical analyses were then performed in Python using the stats package in SciPy.

**Reporting summary**. Further information on research design is available in the Nature Research Reporting Summary linked to this article.

## Data availability

Data sources and permissions are detailed in Supplementary Note 3. Datasets used in these analyses are available publicly, except Traunstein Large Forest Dynamics Plot, Kellogg Biological Station, and Hawaii arthropod data, which were made available by permission of the data owners and can be requested directly from them. The processed data is available at https://doi.org/10.6084/m9.figshare.20288595.

## Code availability

Code to reproduce the analyses and generate all figures in the text is available at github.com/micbru/equation_of_state/. Data cleaning code is available as R scripts on request from the authors.

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

## Acknowledgements
We acknowledge a large number of scientists, field technicians, and funding agencies responsible for the data analyzed in this article. Detailed acknowledgements are available in Supplementary Note 3. JH thanks the Santa Fe Institute and The Rocky Mountain Biological Laboratory for logistic support and useful conversations. Funding for this research was provided by grant DEB 1753180 to J.H. from the US National Science Foundation.

## Author contributions
Conceptualization: J.H. Development: J.H., M.B., K.U., and E.A.N. Data curation and management: E.A.N. Visualization: M.B. Writing: J.H., M.B., E.A.N., and K.U.

## Competing interests
The authors declare no competing interests.
