## [Peer Review File · Communications Biology]

Reviewers' comments:

Reviewer #1 (Remarks to the Author):

This manuscript extends METE theory from existing theory that involves three macro state variables (S , N , E or richness, community abundance, and community energy flux) to involve a 4th (B or total community biomass). An analogy is made to chemistry's $PV=nRT$ equation which is fair, although the 4-variable relationship here is a good deal more mathematically complex and partitions the four variables into pairs to a greater degree.

This extension occurs by summing a biomass proxy (metabolic rate to the $4/3$ power) over all metabolic rate and abundance bins using METE predictions and then making a number of approximations. This is tested by comparing predicted vs observed B with $R^2=0.994$ given E , N , S which generally has to count as a highly successful prediction in ecology.

My biggest question is how much of an advance adding B is. It basically takes the metabolic scaling rule relating biomass and energy flux at the individual level ($e=m^{(3/4)}$) and develops a new, much more complex relation between the same two quantities B and E at the total community level, also bringing in N and S . Bringing in N is to my mind not surprising for an equation that seeks to go from the individual (of which there are N units) to the total community level. Bringing in S is to my mind the more interesting piece. It provides some obvious commentary on the whole BEF (biodiversity Ecosystem Function) research program although these conceptual links are not made in this manuscript. And others have grappled with finding the right scaling from individual b , e to community B , E in some high profile papers (thinking e.g. of some papers in Nature and Science by Michaletz, Enquist and others) so this might be a valuable contribution.

A related concern is the testing. How many of the datasets actually measure total community energy flux and total community biomass independently? VS how many are calculating one or both by measuring individual biomass, calculating individual metabolic rate (by scaling relation) and then summing up to get total community flux. This is probably the most common way to estimate total community biomass & flux. Weighing whole communities (e.g. via destructive harvesting) or measuring whole community energy flux (e.g. via FluxNet towers) is relatively rare. And if most or all data sets are measured at the individual level and then summed to get total community values, how much is matching an equation that is mathematically based on summing from the individual level an independent test vs tautological (at that point only the approximations are being tested and they are mathematical laws). Moreover how many papers measure even at the individual level both mass, m , and energy flux, ϵ vs using the scaling rule (again just matching the theory in the measurement). Much more detail on how E and B are measured in the test datasets is needed to convince me of the degree to which the theory is truly independent of rather than just echoing the measurement methods.

A couple of small points:

1) the approximation depends on $1 \ll S \ll N \ll E$. S and N are counts so an intuitive comparison to make and this relation is fairly well discussed and usually holds. The assumption $E \gg N$ is more of a stretch as it would clearly depend on units of E , and the somewhat unusual units used of e_{\min} (energy flux of smallest individual)=1 is used. I think if one then argues that energy fluxes are lognormally (or at least highly right skewed on an arithmetic axis) distributed one can get to a convincing argument that $E \gg N$ in these weird units, but I've never seen such an argument so a little effort is needed.

2) Equation 3 is hardly easily interpretable. it would help me to see some simple plots of B vs E holding S & N constant, and so on across all possible pairs of state variables.

Otherwise, to my reading the manuscript is correct and clear. It's certainly a nifty result. How general the interest will be is ultimately something for the editor to determine.

Reviewer #2 (Remarks to the Author):

In this manuscript, Harte et al. attempt to derive an equation of state among species richness, energy flow, biomass, and abundance, combining METE and MTE, akin to equations of state in thermodynamics, such as the ideal gas law.

On the positive side, the manuscript is well-written, although a bit jargon-laden. The proposal for integrating METE and MTE is important and a necessary step for a more comprehensive theory of ecology.

On the negative side, there are several problems with this manuscript. First, as mentioned by the authors, the state variables in METE are assumed to be independent. However, as shown in the literature and the Appendix, and at odds with the author's own claim, these are not independent. The authors explain that a strong correlation between B and E is expected as they are computed from the same data, but several state variables are not independent. The existence of correlations and interdependence among state variables is an important problem, specially since most state variables can be derived from each other, as it is between N and S (e.g., Fisher's diversity) and B and S (see paper by Enquist and Niklas 2001) and B and E (Enquist et al. 2007).

Other scientists have provided theoretical arguments for the existence of covariation in several scaling laws (Gaston and Blackburn 2001, Zaoli et al. 2017) linked to resource supply rates for example, that need to be cited in the introduction.

Enquist, B. J., & Niklas, K. J. (2001). Invariant scaling relations across tree-dominated communities. *Nature*, 410(6829), 655-660.

Enquist, B. J., Kerkhoff, A. J., Huxman, T. E., & Economo, E. P. (2007). Adaptive differences in plant physiology and ecosystem paradoxes: insights from metabolic scaling theory. *Global Change Biology*, 13(3), 591-609.

Blackburn, T. M., & Gaston, K. J. (2001). Linking patterns in macroecology. *Journal of animal ecology*, 70(2), 338-352.

Zaoli, S., Giometto, A., Maritan, A., & Rinaldo, A. (2017). Covariations in ecological scaling laws fostered by community dynamics. *Proceedings of the National Academy of Sciences*, 114(40), 10672-10677.

Revisions of COMMSBIO-22-0745-T in Response to Reviewers.

The reviewer comments were constructive. We have now made a number of changes in the manuscript in response and wish to thank the reviewers for their insights. Below we list, point by point, the changes we have made.

Reviewer 1.

This manuscript extends METE theory from existing theory that involves three macro state variables (S , N , E or richness, community abundance, and community energy flux) to involve a 4th (B or total community biomass). An analogy is made to chemistry's $PV=nRT$ equation which is fair, although the 4-variable relationship here is a good deal more mathematically complex and partitions the four variables into pairs to a greater degree.

This extension occurs by summing a biomass proxy (metabolic rate to the $4/3$ power) over all metabolic rate and abundance bins using METE predictions and then making a number of approximations.

To clarify, the graphs showing the test of the Equation of State, evaluate numerically the predicted B from S , N , and E *without approximations*. We have made this more clear in the text (p. 5, lines 13-15). Approximations are only used to derive an accurate but not exact closed-form analytical expression for the equation of state.

This is tested by comparing predicted vs observed B with $R^2=0.994$ given E , N , S which generally has to count as a highly successful prediction in ecology.

Thank you.

My biggest question is how much of an advance adding B is. It basically takes the metabolic scaling rule relating biomass and energy flux at the individual level ($e=m^{3/4}$) and develops a new, much more complex relation between the same two quantities B and E at the total community level, also bringing in N and S . Bringing in N is to my mind not surprising for an equation that seeks to go from the individual (of which there are N units) to the total community level. Bringing in S is to my mind the more interesting piece. It provides some obvious commentary on the whole BEF (biodiversity Ecosystem Function) research program although these conceptual links are not made in this manuscript.

An excellent point. We have now made that link to the BEF literature (p. 6, lines 25, 26 and ensuing discussion).

And others have grappled with finding the right scaling from individual b , e to community B , E in some high profile papers (thinking e.g. of some papers in Nature and Science by Michaletz, Enquist and others) so this might be a valuable contribution.

We have added discussion of, and citations to, previous work by others on this and related scaling questions (p. 4, lines 6 - 10)

A related concern is the testing. How many of the datasets actually measure total community energy flux and total community biomass independently? VS how many are calculating one or both by measuring individual biomass, calculating individual metabolic rate (by scaling relation) and then summing up to get total community flux. This is probably the most common way to estimate total community biomass & flux. Weighing whole communities (e.g. via destructive harvesting) or measuring whole community energy flux (e.g. via FluxNet towers) is relatively rare. And if most or all data sets are measured at the individual level and then summed to get total community values, how much is matching an equation that is mathematically based on summing from the individual level an independent test vs tautological (at that point only the approximations are being tested

and they are mathematical laws). Moreover how many papers measure even at the individual level both mass, m , and energy flux, ϵ vs using the scaling rule (again just matching the theory in the measurement). Much more detail on how E and B are measured in the test datasets is needed to convince me of the degree to which the theory is truly independent of rather than just echoing the measurement methods.

In Data Sources in Supplementary Material, we describe the methods used in the various data sets to estimate biomass and metabolic rate. In no case are separate measurements used to estimate B and E . The question raised by the reviewer as to whether this then leads to tautology is now addressed in the main text (p. 5, lines 25-30; p. 6, lines 1, 2). To summarize the main point, given a single set of measurements, of either individual masses or individual metabolic rates, application of the metabolic scaling rule relating these, and then summing over individuals, leads only to an inequality between $B^{3/4}$ and E (p. 5, line 30). We have now made clear that in our equation of state, the predicted function of S and N in Eq. 3 is actually a prediction for the strength of that inequality (p. 6, lines 1,2; Fig. 3).

A couple of small points:

1) the approximation depends on $1 \ll s \ll n \ll N$ is more of a stretch as it would clearly depend on units of E , and the somewhat unusual units used of e_{\min} (energy flux of smallest individual)=1 is used. I think if one then argues that energy fluxes are lognormally (or at least highly right skewed on an arithmetic axis) distributed one can get to a convincing argument that $E \gg N$ in these weird units, but I've never seen such an argument so a little effort is needed.

The choice of units affects the numerical values of the upper and lower bounds on the integrals in METE. And that choice translates into different inequality requirements and different values of state variables and Lagrange multipliers, just as in any application of units conversion. But the validity of the equation of state and of the other predictions of METE is independent of the units, and that is why we work consistently with a single, albeit arbitrary, choice of units. We cite (p. 4, line 22) a publication that delves more deeply into this.

2) Equation 3 is hardly easily interpretable. it would help me to see some simple plots of B vs E holding S & N constant, and so on across all possible pairs of state variables.

This is a very useful suggestion. We agree that it is not easy to get an intuitive sense of the role of S and N in the equation of state. To remedy that, we have added Figure 3, and accompanying discussion (p. 6, lines 19 - 23), to describe the predicted dependence of the ratio $E/B^{3/4}$ on S and N .

Otherwise, to my reading the manuscript is correct and clear. It's certainly a nifty result. How general the interest will be is ultimately something for the editor to determine.

Thank you.

Reviewer 2.

In this manuscript, Harte et al. attempt to derive an equation of state among species richness, energy flow, biomass, and abundance, combining METE and MTE, akin to equations of state in thermodynamics, such as the ideal gas law.

On the positive side, the manuscript is well-written, although a bit jargon-laden. The proposal for integrating METE and MTE is important and a necessary step for a more comprehensive theory of ecology.

We have tried to make this paper accessible to scientists working in diverse disciplines, and thus to eliminate unnecessary jargon. Where we have used terms like “state variable”, “micro” or “macro” we have tried to explain them either explicitly or by their context.

On the negative side, there are several problems with this manuscript. First, as mentioned by the authors, the state variables in METE are assumed to be independent. However, as shown in the literature and the Appendix, and at odds with the author’s own claim, these are not independent.

The authors explain that a strong correlation between B and E is expected as they are computed from the same data, but several state variables are not independent. The existence of correlations and interdependence among state variables is an important problem, specially since most state variables can be derived from each other, as it is between N and S (e.g., Fisher’s diversity) and B and S (see paper by Enquist and Niklas 2001) and B and E (Enquist et al. 2007).

Other scientists have provided theoretical arguments for the existence of covariation in several scaling laws (Gaston and Blackburn 2001, Zaoli et al. 2017) linked to resource supply rates for example, that need to be cited in the introduction.

The reviewer is correct about this: the state variables are not uncorrelated across ecosystems. The wording in our original ms, in which we stated that the state variables, S , N and E , were independent of each other, was imprecise. What we intended, and have now made clear in the revision (p. 3, lines 26-31; p. 4, lines 1-4) is that no one of those three variables can be *accurately* predicted by another or by the other two. This was always evident from Appendix 1, which shows weak correlation (considerable scatter) in the pairwise graphs between these variables, and from the Supplementary Video, which shows the same in 3-D graphs of all three variables. We thank the reviewer for catching this.

And the subsequent points made by the reviewer are also correct. See the 4th and 5th responses to reviewer 1, which address exactly these issues. In addition to the revisions listed above, we have made explicit how our work relates to other efforts, including the Fisher’s “alpha” measure of diversity (p. 3, lines 26-31; p. 4, lines 1-13), and also added citations to more of the literature on relationships among state variables and on covariation in scaling laws as requested (new citations 5-8, 19-21, 42, 43).

REVIEWERS' COMMENTS:

Reviewer #1 (Remarks to the Author):

I think this is an interesting, potentially important manuscript. I feel like the responses to comments was good and it is now a tight manuscript. I especially found Figure 3 a nice addition. I look forward to being able to cite this manuscript.

My only remaining (small) comment is on the topic of the approximation and when it is relevant. Line 136 effectively says the approximation only applies when the average abundance of a species in the hundreds and the average metabolic rate of an individual is >50 times the metabolic rate of the smallest (lowest metabolic rate) organism in the community. This still seems to me like a stretch that applies to relatively few ecological data sets, unless I am missing something. That said I do recognize most of the paper relies on the precise equation 1 rather than the approximate equation 2 and that there is still use in exploring asymptotic boundaries even if they aren't usually achieved.

Reviewer #2 (Remarks to the Author):

I commend the authors for this new version that addresses my comments fully. This is an important paper that will nicely help advance the integration of theories' agenda.

Response to Reviewer Comment

A reviewer made the following excellent point:

"My only remaining (small) comment is on the topic of the approximation and when it is relevant. Line 136 effectively says the approximation only applies when the average abundance of a species in the hundreds and the average metabolic rate of an individual is >50 times the metabolic rate of the smallest (lowest metabolic rate) organism in the community. This still seems to me like a stretch that applies to relatively few ecological data sets, unless I am missing something. That said I do recognize most of the paper relies on the precise equation 1 rather than the approximate equation 2 and that there is still use in exploring asymptotic boundaries even if they aren't usually achieved."

We agree and have inserted into the text (p. 5, line 25 to p. 6, line 3) explicit mention of the fact that a number of ecosystems possess state variables that do not satisfy the bounds required if either or both of the two analytical approximations that we present are to be valid. We also make this quantitative by providing the fractions (5/42 and 39/42) of the data sets we analyzed for which the stronger and weaker bounds (respectively) *do* apply. We emphasize, as did the reviewer, that the empirical tests of our prediction presented in the paper do not rely on the validity of either approximation.

The other changes in our manuscript involve formatting issues and the transfer of some material in the Supplementary Information to a new Methods Section.